# Racial disparities in the SOFA score among patients hospitalized with COVID-19

**Benjamin Tolchin**[1]*, **Carol Oladele**[2], **Deron Galusha**[2], **Nitu Kashyap**[3], **Mary Showstark**[4], **Jennifer Bonito**[5], **Michelle C. Salazar**[6], **Jennifer L. Herbst**[7], **Steve Martino**[8], **Nancy Kim**[9], **Katherine A. Nash**[10], **Max Jordan Nguemeni Tiako**[2], **Shireen Roy**[11], **Rebeca Vergara Greeno**[2], **Karen Jubanyik**[5]

1 Department of Neurology, Yale School of Medicine, New Haven, Connecticut, United States of America, 2 Equity Research and Innovation Center, Yale School of Medicine, New Haven, Connecticut, United States of America, 3 Information Technology, Yale New Haven Health, New Haven, Connecticut, United States of America, 4 Yale School of Medicine Physician Assistant Online Program, New Haven, Connecticut, United States of America, 5 Department of Emergency Medicine, Yale School of Medicine, New Haven, Connecticut, United States of America, 6 Department of Surgery, Yale School of Medicine, New Haven, Connecticut, United States of America, 7 Quinnipiac University School of Law, North Haven, Connecticut, United States of America, 8 Department of Psychiatry, Yale University School of Medicine, New Haven, Connecticut, United States of America, 9 Department of Medicine, Yale School of Medicine, New Haven, Connecticut, United States of America, 10 Department of Pediatrics, Yale School of Medicine, New Haven, Connecticut, United States of America, 11 Washington University School of Medicine in St. Louis, St. Louis, Missouri, United States of America

* benjamin.tolchin@yale.edu

**Data Availability Statement:** We have discussed the sharing of our data with the Yale University Privacy Office which made the determination that we are legally and ethically restricted from sharing

## Abstract

### Background

Sequential Organ Failure Assessment (SOFA) score predicts probability of in-hospital mortality. Many crisis standards of care suggest the use of SOFA scores to allocate medical resources during the COVID-19 pandemic.

### Research question

Are SOFA scores elevated among Non-Hispanic Black and Hispanic patients hospitalized with COVID-19, compared to Non-Hispanic White patients?

### Study design and methods

Retrospective cohort study conducted in Yale New Haven Health System, including 5 hospitals with total of 2681 beds. Study population drawn from consecutive patients aged $\geq$18 admitted with COVID-19 from March 29$^{th}$ to August 1$^{st}$, 2020. Patients excluded from the analysis if not their first admission with COVID-19, if they did not have SOFA score recorded within 24 hours of admission, if race and ethnicity data were not Non-Hispanic Black, Non-Hispanic White, or Hispanic, or if they had other missing data. The primary outcome was SOFA score, with peak score within 24 hours of admission dichotomized as <6 or $\geq$6.

data because the extent of data poses a risk of re-identification of patients and their HIPAA protected data through deductive disclosure. Susan Bouregy, PhD (susan.buregy@yale.edu) is Yale's chief privacy officer, and will serve as the contact for the Yale University Privacy Office, to which data requests may be sent.

**Funding:** The authors received no specific funding for this work. BT receives research support from the US Department of Veterans Affairs (https://www.newengland.va.gov/research/v1cda/) and the C.G. Swebilius Foundation (https://fconline.foundationcenter.org/fdo-grantmaker-profile/?key=SWEB001). KN and MS are supported by the National Clinician Scholars Program and the CTSA Grant Number TL1 TR001864 from the National Center for Advancing Translational Science (NCATS), a component of the National Institutes of Health (NIH). (https://ncats.nih.gov/) The manuscript's contents are solely the responsibility of the authors and do not necessarily represent the official view of the NIH. The funders had no role in study design, data collection and analysis, decision to publish, or preparation of the manuscript.

**Competing interests:** The authors have declared that no competing interests exist.

## Results

Of 2982 patients admitted with COVID-19, 2320 met inclusion criteria and were analyzed, of whom 1058 (45.6%) were Non-Hispanic White, 645 (27.8%) were Hispanic, and 617 (26.6%) were Non-Hispanic Black. Median age was 65.0 and 1226 (52.8%) were female. In univariate logistic screen and in full multivariate model, Non-Hispanic Black patients but not Hispanic patients had greater odds of an elevated SOFA score ≥6 when compared to Non-Hispanic White patients (OR 1.49, 95%CI 1.11–1.99).

## Interpretation

Given current unequal patterns in social determinants of health, US crisis standards of care utilizing the SOFA score to allocate medical resources would be more likely to deny these resources to Non-Hispanic Black patients.

## Introduction

Prior to the first wave of Coronavirus-2019 (COVID-19), models predicted that a pandemic respiratory virus might require ventilators, intensive care unit (ICU) beds, and other life-sustaining medical resources far in excess of available supplies [1]. On January 30th 2020, the World Health Organization (WHO) declared a Public Health Emergency of International Concern which, in some countries, early on led to formal and informal restrictions on the allocation of critical medical resources on the basis of advanced age [2, 3]. These policies and recommendations were quickly criticized within their home countries and revised to promote equal access to care, but nonetheless informed the development of early COVID-19 crisis standards of care (CSC) in the United States (US) [4].

In response to early shortages and high rates of infection and mortality in Europe and the Northeastern US, a number of healthcare systems and states in the US developed CSC: guidelines that advise hospitals and providers how to operate in a public health disaster, outside of their normal operating standards of care. CSC include triage protocols for the allocation of scarce life-sustaining medical resources [5–9]. The primary goal of published protocols was to establish a consistent system for allocating resources to save as many lives as possible during public health emergencies. A potential alternative goal of saving as many *life-years* as possible was widely rejected as being likely to unjustly discriminate against marginalized racial and ethnic groups, people with disabilities, people of advanced age, and others with a shorter long-term life expectancy [10].

Most publicly available US triage protocols, prior to and during the pandemic, used the Sequential Organ Failure Assessment (SOFA) score, with or without additional prognostic factors, to assess patients' likelihood of benefiting (surviving) as a result of receiving medical resources [9, 11]. The rationale was that if a severe shortage of critical medical resources did occur, then the limited resources would be directed to those most likely to survive as a result of receiving them, thereby saving the most lives possible. Without standardized protocols the allocation of scarce resources is likely to be highly variable and inequitable. The SOFA score is a validated prognostic score ranging from 0–24, with points assigned for evidence of organ failure within 6 different organ systems, with higher scores correlating with a higher likelihood of in-hospital mortality [12, 13]. Originally developed and validated among septic patients in the medical ICU, subsequent research during the COVID pandemic has shown that the SOFA

score is actually poorly predictive of mortality among patients with acute respiratory distress syndrome (ARDS) in the setting of COVID-19 infection; it is less accurate than either the Acute Physiology and Chronic Health Evaluation (APACHE) II score or chronologic age [14, 15]. Triage protocols utilizing the SOFA score were not widespread in Europe and even in the US these CSC were generally not actually activated during the pandemic. Nonetheless, most published US disaster triage protocols prioritized patients who require medical resources but have lower SOFA scores to receive resources, on the grounds that such patients are more likely to benefit (survive), and there is concern that these protocols will guide medical decision making in the US in the event of future public health emergencies [9].

In addition to threatening to overwhelm existing medical resources, the COVID-19 pandemic has also highlighted and exacerbated existing racial, ethnic, and socioeconomic health disparities. Marginalized populations, including racial and ethnic minorities and individuals of lower socioeconomic status, are more likely to become infected with COVID-19, more likely to be hospitalized, and more likely to die as a result [16–18]. Disparities in social determinants of health, including safe access to adequate nutritious food, exercise options, stable housing, and economic opportunities likely contribute to disparities in COVID-19 outcomes.

Marginalized populations are more likely to work in service-sector jobs that cannot be conducted remotely, are more likely to depend on public transportation, and are more likely to live in small and densely packed housing units and in group-living situations including homeless shelters, prisons, jails, and detention facilities [19–22]. They are less likely to have access to preventive healthcare and more likely to experience bias when they do access the healthcare system, resulting in higher rates of chronic comorbidities including diabetes, hypertension, and chronic pulmonary diseases [23, 24]. These pervasive inequalities in social determinants of health, and the health inequities that they cause, constitute a structure of systemic racism and contribute to higher rates of COVID-19 infection, more severe acute illness due to preexisting conditions, and higher mortality rates [25, 26].

Given that marginalized populations are more likely to become sicker with COVID-19, utilization of CSC triage protocols, which rely on the SOFA score, have the potential to disproportionately deny medical resources to racial and ethnic minorities [27, 28]. Racial, ethnic, and socioeconomic inequities in health outcomes are a consequence of inequalities in social determinants of health, and they can potentially be exacerbated by triage protocols, as documented in patient cohorts with sepsis and ARDS but not previously examined in patients with COVID-19 [29]. The use of SOFA as a criterion to allocate scarce medical resources has the potential to exacerbate inequities caused by social determinants of health. We conducted a retrospective cohort study to assess whether SOFA scores are disproportionately elevated among members of racial and ethnic minorities, and specifically Non-Hispanic Black and Hispanic patients, in comparison to Non-Hispanic White patients with COVID-19. The existence of such a disparity would raise significant concerns about the use of triage protocols relying on SOFA scores and the potential for exacerbating racial and ethnic health inequities during future waves of the COVID-19 pandemic and other public health emergencies.

## Methods

### Study design and data source

We conducted a retrospective cohort study of patients with COVID-19 within the Yale-New Haven Health System (YNHH) from March 29th, 2020 to August 1, 2020. YNHH includes 5 hospitals and a large physician practice base, serving racially, ethnically, and socioeconomically diverse communities across Connecticut and Rhode Island. The hospitals range from primary community hospitals to a tertiary academic medical center, with a total of 2,681 beds. During

the pandemic, the healthcare system worked as a unified network, with sicker patients transferred from smaller hospitals within the network to Yale New Haven Hospital, all operating under a single protocol. Across the system, patients are 61% White, 17% Black, 3% Asian, 0.3% Native American, 0.3% Hawaiian or other Pacific Islander, and 19% other or unknown race. Patients are 78% Non-Hispanic, 20% Hispanic, and 2% other or unknown ethnicity. Forty-six percent of patients have primary private insurance, 19% have primary Medicare, 28% have primary Medicaid, and 7% are uninsured.

Connecticut and Rhode Island Medicaid cover low-income residents who are US nationals, citizens, permanent residents, or legal aliens [30, 31] Emergency Medicaid, available to all residents regardless of immigration status, during the COVID-19 public health emergency was expanded to include COVID-19 treatment and hospitalization [32, 33]. Personal asset and income limits vary across states and within states and are available [30, 31, 34]. Generally, Medicaid enrollees in Connecticut and Rhode Island do not pay premiums, deductibles, or copayments [34].

Data from the YNHH electronic medical record (EMR, Epic Systems Corporation, Verona, WI) database was used for analyses. The study was approved by the Yale University Human Subjects Committee (study number 2000028081), which judged that the study was exempt from the requirement for consent because data were analyzed anonymously.

## Participants

We included EMR data for all patients age ≥18 with COVID-19 admitted to YNHH hospitals during the study period. Patients were considered positive for COVID-19 if they had a positive PCR test or clinical markers including fever, cough and chest radiographs considered to be consistent with COVID-19 infection in the setting of the first wave of the pandemic in the northeastern United States, and designated as COVID-19 positive by an attending physician. Patients <18 years of age were excluded as the SOFA score is not validated in pediatric patients. Patients were excluded from the analysis if they did not have a SOFA score recorded within 24 hours of admission, if it was not their first admission with COVID-19, or if their race and ethnicity data were not Non-Hispanic Black, Non-Hispanic White, or Hispanic (Fig 1). While YNHH serves significant Hispanic and Non-Hispanic Black patient populations, it serves relatively smaller numbers of Asian, Native American, Pacific Islander, and other patient populations, and these small samples statistically prevented inclusion in the analysis. Because prior COVID-19 studies show that Black and Hispanic patients experience higher rates of critical illness and mortality [16–18], we hypothesized that Non-Hispanic Black and Hispanic patients will be more likely to have elevated SOFA scores within 24 hours of admission compared to Non-Hispanic White patients.

## Predictor variables

Data extracted from the EMR included sociodemographic and clinical variables. Our main predictor variables were age, sex, race, ethnicity, and insurance status. These variables are recorded by admitting clerks at YNHH hospitals. Other variables included clinical characteristics like body mass index (BMI) and comorbid conditions known to be associated with mortality in the setting of COVID-19 and accurately captured in the electronic medical record (including chronic pulmonary disease, congestive heart failure, diabetes, coronary artery disease, hypertension, advanced renal disease, and advanced liver disease) [18]. Smoking status was not included in the analysis, because in our clinical experience there is a significant desirability bias, leading patients to report themselves to clinicians as non-smokers or former smokers rather than current smokers [35].

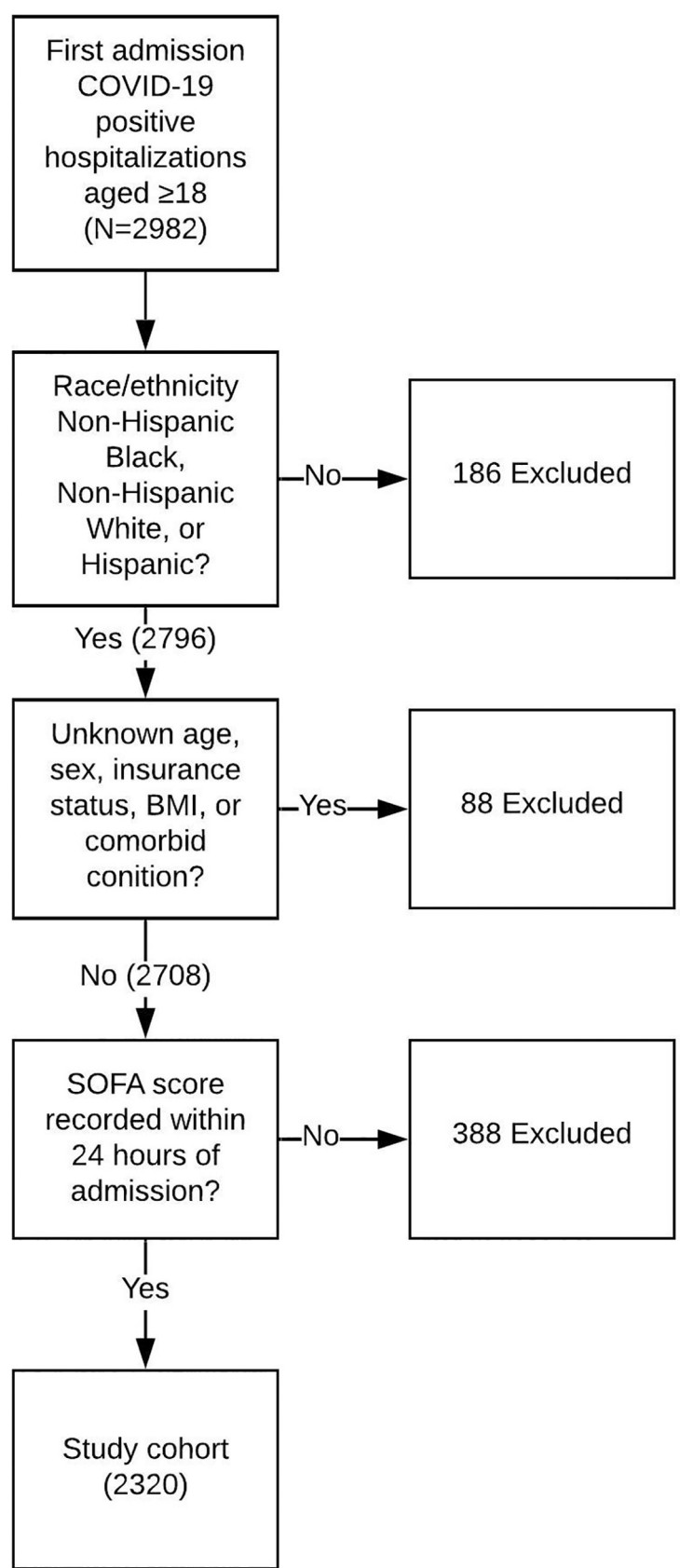

**Fig 1. Construction of study cohort.** Abbreviations: BMI: Body Mass Index; COVID-19: Coronavirus Disease 2019; SOFA: Sequential Organ Failure Assessment.

## Outcome variable

The main outcome, SOFA score, was continuously and automatically calculated for all admitted patients and recorded every 4 hours within the YNHH EMR. Triage decisions were never made at YNHH on the basis of this calculated SOFA score, because the Triage Protocol was approved but never activated during the pandemic [7]. SOFA scores were determined by an automated algorithm within the EMR system, assigning 0–4 points for each of 6 organ systems (neurologic, pulmonary, cardiovascular, renal, hepatic, hematologic), based on laboratory, respiratory and nursing flowsheet data in the EMR, following previously specified and validated rules [12]. The total SOFA score ranges from 0–24, with higher scores indicating a higher likelihood of in-hospital mortality. A binary SOFA variable (peak score within 24 hours <6, ≥6) was created to examine variation in illness severity by patient sociodemographic characteristics. We focused on this dichotomous outcome because published triage protocols categorize patients with a SOFA score <6 as being in the most prioritized group, most likely to receive scarce medical resources in a disaster situation, whereas patients with SOFA score ≥6 are deprioritized, resulting in lower likelihoods of receiving scarce medical resources [5, 7]. We focused on peak SOFA score within the first 24 hours because in a public health emergency in which life-sustaining medical resources are fully occupied, it is initial SOFA scores that will determine whether a newly admitted critically ill patient receives scarce resources.

## Statistical analysis

We used Analysis of Variance (ANOVA) tests to examine mean differences in peak 24-hour SOFA score by sociodemographic and clinical characteristics. Chi-square tests were used to examine differences in the proportion of COVID positive patients with SOFA score ≥6 and <6 by patient characteristics. Finally, we conducted logistic regression analyses to assess racial differences in SOFA score adjusting for sociodemographic and clinical covariates. We considered candidate covariates based on clinical experience and emerging evidence regarding associations with clinical outcomes in COVID-19. The final multivariate model was then refined through the exclusion of collinear covariates. We conducted a univariate screen followed by a multivariate regression adjusting for all sociodemographic and clinical covariates listed in Table 4 (race/ethnicity, age, sex, BMI, insurance status, comorbid conditions including chronic pulmonary disease, congestive heart failure, diabetes, coronary artery disease, hypertension, advanced renal disease, and advanced liver disease). Race-stratified models were also constructed to assess whether factors associated with SOFA score varied according to race and ethnicity. We also conducted a multiple mixed-effects logistic regression, including random intercepts placed on hospital of admission to account for potential inter-hospital variability.

## Results

From March 29[th] to August 1[st], there were 3362 admissions of COVID-19-positive patients aged ≥18 to YNHH hospitals. Of these, 2982 were first admissions (Fig 1) and 2796 were Non-Hispanic Black, Non-Hispanic White, or Hispanic. Of these, 88 had missing baseline demographics or clinical data, and 388 had missing SOFA scores, and were excluded. Two thousand three hundred and twenty patients had complete race/ethnicity and baseline characteristics, were either Hispanic, Non-Hispanic Black, or Non-Hispanic White, and were included in the

**Table 1. Characteristics of COVID+ patients with SOFA within 24 hours of admission; SOFA <6 and ≥6.**

| Characteristic | Total (n = 2,320) | | 24-hour Sofa < 6 (n = 1,985) | | 24-hour Sofa ≥ 6 (n = 335) | | p-value |
|---|---|---|---|---|---|---|---|
| | n | % | n | % | n | % | |
| **Race/Ethnicity** | | | | | | | < .0001 |
| Hispanic | 645 | 27.8 | 575 | 29.0 | 70 | 20.9 | |
| Black Non-Hispanic | 617 | 26.6 | 495 | 24.9 | 122 | 36.4 | |
| White Non-Hispanic | 1058 | 45.6 | 915 | 46.1 | 143 | 42.7 | |
| **Age** | | | | | | | 0.0008 |
| 18–34 | 224 | 9.7 | 210 | 10.6 | 14 | 4.2 | |
| 35–64 | 895 | 38.6 | 765 | 38.5 | 130 | 38.8 | |
| > = 65 | 1201 | 51.8 | 1010 | 50.9 | 191 | 57.0 | |
| **Sex** | | | | | | | < .0001 |
| Men | 1094 | 47.2 | 893 | 45.0 | 201 | 60.0 | |
| Women | 1226 | 52.8 | 1092 | 55.0 | 134 | 40.0 | |
| **Language preference** | | | | | | | 0.2679 |
| English | 1850 | 79.7 | 1572 | 79.2 | 278 | 83.0 | |
| Spanish | 418 | 18.0 | 368 | 18.5 | 50 | 14.9 | |
| Other | 52 | 2.2 | 45 | 2.3 | 7 | 2.1 | |
| **Insurance status** | | | | | | | 0.0472 |
| Private | 434 | 18.7 | 386 | 19.4 | 48 | 14.3 | |
| Medicare | 1227 | 52.9 | 1031 | 51.9 | 196 | 58.5 | |
| Medicaid | 480 | 20.7 | 409 | 20.6 | 71 | 21.2 | |
| Uninsured | 179 | 7.7 | 159 | 8.0 | 20 | 6.0 | |
| **BMI** | | | | | | | 0.8194 |
| <25 | 677 | 29.2 | 585 | 29.5 | 92 | 27.5 | |
| 25–29.9 | 674 | 29.1 | 577 | 29.1 | 97 | 29.0 | |
| 30–34.9 | 455 | 19.6 | 389 | 19.6 | 66 | 19.7 | |
| 35+ | 514 | 22.2 | 434 | 21.9 | 80 | 23.9 | |
| **Comorbid conditions** | | | | | | | |
| Chronic pulmonary disease | 702 | 30.3 | 586 | 29.5 | 116 | 34.6 | 0.0599 |
| CHF | 573 | 24.7 | 454 | 22.9 | 119 | 35.5 | < .0001 |
| Diabetes | 1001 | 43.1 | 821 | 41.4 | 180 | 53.7 | < .0001 |
| CAD | 594 | 25.6 | 470 | 23.7 | 124 | 37.0 | < .0001 |
| Hypertension | 1601 | 69.0 | 1341 | 67.6 | 260 | 77.6 | 0.0002 |
| Advance renal disease | 206 | 8.9 | 141 | 7.1 | 65 | 19.4 | < .0001 |
| Advance liver disease | 46 | 2.0 | 31 | 1.6 | 15 | 4.5 | 0.0004 |

Abbreviations: BMI: Body Mass Index; CAD: Coronary Artery Disease; CHF: Congestive Heart Failure; SOFA: Sequential Organ Failure Assessment

analysis. There were no statistically significant differences in demographic or clinical characteristics between patients with and without SOFA scores.

Within the study cohort of 2320, 1058 (45.6%) were Non-Hispanic White, 645 (27.8%) were Hispanic, and 617 (26.6%) were Non-Hispanic Black (Table 1). The median age was 65.0, and 1226 (52.8%) were female. Six-hundred and fifty-nine (28.4%) had Medicaid or no insurance. Nine-hundred and sixty-nine (41.7%) were obese. A total of 1829 (78.8%) had one or more comorbid conditions thought to increase risk of mortality in the setting of COVID-19. Patients with peak SOFA scores ≥6 within the first 24 hours were disproportionately common among Non-Hispanic Black patients, older patients, males, and patients with Congestive Heart Failure (CHF), diabetes, coronary artery disease (CAD), hypertension, advanced renal disease,

**Table 2. Characteristics of COVID+ patients by race/ethnicity.**

| Characteristic | Total (n = 2,320) | | Hispanic (n = 645) | | Non-Hispanic Black (n = 617) | | Non-Hispanic White (n = 1,058) | | p-value |
|---|---|---|---|---|---|---|---|---|---|
| | n | % | n | % | n | % | N | % | |
| **Age** | | | | | | | | | < .0001 |
| 18–34 | 224 | 9.7 | 121 | 18.8 | 58 | 9.4 | 45 | 4.3 | |
| 35–64 | 895 | 38.6 | 344 | 53.3 | 284 | 46.0 | 267 | 25.2 | |
| > = 65 | 1201 | 51.8 | 180 | 27.9 | 275 | 44.6 | 746 | 70.5 | |
| **Sex** | | | | | | | | | 0.0002 |
| Men | 1094 | 47.2 | 348 | 54.0 | 268 | 43.4 | 478 | 45.2 | |
| Women | 1226 | 52.8 | 297 | 46.0 | 349 | 56.6 | 580 | 54.8 | |
| **Language preference** | | | | | | | | | < .0001 |
| English | 1850 | 79.7 | 226 | 35.0 | 597 | 96.8 | 1027 | 97.1 | |
| Spanish | 418 | 18.0 | 415 | 64.3 | 0 | 0.0 | 3 | 0.3 | |
| Other | 52 | 2.2 | 4 | 0.6 | 20 | 3.2 | 28 | 2.6 | |
| **Insurance status** | | | | | | | | | < .0001 |
| Private | 434 | 18.7 | 116 | 18.0 | 134 | 21.7 | 184 | 17.4 | |
| Medicare | 1227 | 52.9 | 166 | 25.7 | 296 | 48.0 | 765 | 72.3 | |
| Medicaid | 480 | 20.7 | 224 | 34.7 | 158 | 25.6 | 98 | 9.3 | |
| Uninsured | 179 | 7.7 | 139 | 21.6 | 29 | 4.7 | 11 | 1.0 | |
| **BMI** | | | | | | | | | < .0001 |
| <25 | 677 | 29.2 | 153 | 23.7 | 151 | 24.5 | 373 | 35.3 | |
| 25–29.9 | 674 | 29.1 | 211 | 32.7 | 148 | 24.0 | 315 | 29.8 | |
| 30–34.9 | 455 | 19.6 | 152 | 23.6 | 129 | 20.9 | 174 | 16.4 | |
| 35+ | 514 | 22.2 | 129 | 20.0 | 189 | 30.6 | 196 | 18.5 | |
| **Comorbid conditions** | | | | | | | | | |
| Chronic pulmonary disease | 702 | 30.3 | 137 | 21.2 | 222 | 36.0 | 343 | 32.4 | < .0001 |
| CHF | 573 | 24.7 | 68 | 10.5 | 177 | 28.7 | 328 | 31.0 | < .0001 |
| Diabetes | 1001 | 43.1 | 230 | 35.7 | 354 | 57.4 | 417 | 39.4 | < .0001 |
| CAD | 594 | 25.6 | 81 | 12.6 | 171 | 27.7 | 342 | 32.3 | < .0001 |
| Hypertension | 1601 | 69.0 | 294 | 45.6 | 500 | 81.0 | 807 | 76.3 | < .0001 |
| Advance renal disease | 206 | 8.9 | 27 | 4.2 | 100 | 16.2 | 79 | 7.5 | < .0001 |
| Advance liver disease | 46 | 2.0 | 15 | 2.3 | 14 | 2.3 | 17 | 1.6 | 0.4917 |

Abbreviations: BMI: Body Mass Index; CAD: Coronary Artery Disease; CHF: Cogestive Heart Failure; SOFA: Sequential Organ Failure Assessment

and advanced liver disease. Baseline characteristics broken down by race/ethnicity are given in Table 2. Non-Hispanic White patients were significantly older than average while Hispanic patients were significantly younger. Non-Hispanic Black patients had higher rates of elevated BMI and most comorbid conditions, including chronic pulmonary disease, diabetes, hypertension, and advanced renal disease.

Mean peak SOFA score within the first 24 hours was (2.4±3.0) overall, ranging from 0 to 18 (Table 3). Mean SOFA score was significantly elevated among Non-Hispanic Black patients (3.0±3.1), but not among Hispanic patients (2.2±3.1) in comparison to Non-Hispanic White patients (2.5±2.8). SOFA score was also significantly elevated among patients aged 35–64 (2.5 ±3.0) and ≥65 (2.8±3.0) in comparison to those aged 18–34 (1.3±2.3), among Men (3.0±3.2) in comparison to Women (2.2±2.6), and among those with Medicare insurance (2.9±3.0) but not Medicaid (2.3±3.0), or no insurance (2.0±3.1) compared to those with private insurance (2.0±2.8). The SOFA score was also significantly elevated among those with comorbid

**Table 3. Mean SOFA within 24 hours of admission.**

| Characteristic | Total | | SOFA within 24 hours | | p-value |
|---|---|---|---|---|---|
| | n | % | Mean | SD | |
| **Race/Ethnicity** | | | | | < .0001 |
| Hispanic | 645 | 27.8 | 2.2 | 3.1 | |
| Black Non-Hispanic | 617 | 26.6 | 3.0 | 3.1 | |
| White Non-Hispanic | 1058 | 45.6 | 2.5 | 2.8 | |
| **Age** | | | | | < .0001 |
| 18–34 | 224 | 9.7 | 1.3 | 2.3 | |
| 35–64 | 895 | 38.6 | 2.5 | 3.0 | |
| > = 65 | 1201 | 51.8 | 2.8 | 3.0 | |
| **Sex** | | | | | < .0001 |
| Men | 1094 | 47.2 | 3.0 | 3.2 | |
| Women | 1226 | 52.8 | 2.2 | 2.6 | |
| **Language preference** | | | | | 0.2923 |
| English | 1850 | 79.7 | 2.6 | 2.9 | |
| Spanish | 418 | 18.0 | 2.3 | 3.3 | |
| Other | 52 | 2.2 | 2.7 | 2.7 | |
| **Insurance status** | | | | | < .0001 |
| Private | 434 | 18.7 | 2.0 | 2.8 | |
| Medicare | 1227 | 52.9 | 2.9 | 3.0 | |
| Medicaid | 480 | 20.7 | 2.3 | 3.0 | |
| Uninsured | 179 | 7.7 | 2.0 | 3.1 | |
| **BMI** | | | | | 0.805 |
| <25 | 677 | 29.2 | 2.6 | 2.8 | |
| 25–29.9 | 674 | 29.1 | 2.6 | 2.9 | |
| 30–34.9 | 455 | 19.6 | 2.4 | 3.1 | |
| 35+ | 514 | 22.2 | 2.6 | 3.1 | |
| **Comorbid conditions** | | | | | |
| **Chronic pulmonary disease** | | | | | 0.086 |
| No | 1618 | 69.7 | 2.5 | 3.0 | |
| Yes | 702 | 30.3 | 2.7 | 3.0 | |
| **CHF** | | | | | < .0001 |
| No | 1747 | 75.3 | 2.3 | 2.8 | |
| Yes | 573 | 24.7 | 3.4 | 3.2 | |
| **Diabetes** | | | | | < .0001 |
| No | 1319 | 56.9 | 2.2 | 2.8 | |
| Yes | 1001 | 43.1 | 3.0 | 3.1 | |
| **CAD** | | | | | < .0001 |
| No | 1726 | 74.4 | 2.3 | 2.8 | |
| Yes | 594 | 25.6 | 3.3 | 3.2 | |
| **Hypertension** | | | | | < .0001 |
| No | 719 | 31.0 | 1.9 | 2.7 | |
| Yes | 1601 | 69.0 | 2.8 | 3.0 | |
| **Advance renal disease** | | | | | < .0001 |
| No | 2114 | 91.1 | 2.3 | 2.9 | |
| Yes | 206 | 8.9 | 4.7 | 3.1 | |
| **Advance liver disease** | | | | | < .0001 |
| No | 2274 | 98.0 | 2.5 | 2.9 | |

(*Continued*)

**Table 3.** (Continued)

| Characteristic | Total | | SOFA within 24 hours | | p-value |
|---|---|---|---|---|---|
| | n | % | Mean | SD | |
| Yes | 46 | 2.0 | 4.8 | 3.9 | |

Abbreviations: BMI: Body Mass Index; CAD: Coronary Artery Disease; CHF: Congestive Heart Failure; SOFA: Sequential Organ Failure Assessment; SD: Standard Deviation.

conditions including CHF (3.4±3.2) compared to those without (2.3±2.8), diabetes (3.0±3.1) compared to those without (2.2±2.8), CAD (3.3±3.2) compared to those without (2.3±2.8), hypertension (2.8±3.0) compared to those without (1.9±2.7), advanced renal disease (4.7±3.1) compared to those without (2.3±2.9), and advanced liver disease (4.8±3.9) compared to those without (2.5±2.9).

In a univariate logistic screen and in a full multivariate model (Table 4), Non-Hispanic Black patients had greater odds of an elevated SOFA score ≥6 when compared to Non-Hispanic White patients (OR 1.49, 95%CI 1.11–1.99). In contrast, Hispanic patients did not have increased odds of an elevated SOFA score. Advanced age was also associated with increased odds of elevated SOFA score (OR 1.95, 95%CI 1.07–3.54 for age 35–64, OR 2.57, 95%CI 1.32–4.98 for age ≥65), as was male sex (OR 1.94, 95%CI 1.51–2.50), body-mass index ≥35 (OR 1.52, 95%CI 1.07–2.18), advanced renal disease (OR 2.35, 95%CI 1.62–3.40), and advanced liver disease (OR 2.51, 95%CI 1.29–4.89). Medicare was associated with increased odds of elevated SOFA score, but dropped out in the multivariate model, when other variables such as age were included. Race stratified models were also constructed but did not identify new covariates associated with elevated SOFA scores in both univariate screen and multivariate logistic analysis. We conducted a multiple mixed-effects logistic regression, including random intercepts placed on hospital of admission to account for potential inter-hospital variability, with unchanged results. We also reran the analysis looking at peak 48 hour SOFA score with unchanged results.

## Discussion

In our cohort of COVID-19 positive patients admitted to YNHH hospitals, Non-Hispanic Black race/ethnicity, male sex, advanced age, stage II or greater obesity, advanced renal disease, and advanced liver disease were all independently associated with significantly higher odds of elevated peak SOFA score ≥6 during the first 24-hours of admission. Hispanic ethnicity was not associated with increased risk of elevated SOFA score. Neither Medicaid, Medicare nor lack of insurance were independently associated with increased odds of elevated SOFA score. Non-Hispanic Black patients were more likely to suffer from chronic comorbidities associated with elevated peak SOFA scores such as obesity and advanced renal disease (Table 2), and their risk of an elevated SOFA score persisted even when such comorbidities and other risk factors were controlled for in the multivariate analysis (Table 4). Neither insurance status nor exposure to chronic comorbidities fully explained the increased risk faced by Non-Hispanic Black patients with COVID-19.

These findings are consistent with prior studies showing that Black race, older age, obesity, and chronic medical comorbidities are associated with increased rates of mortality in COVID-19 [18]. These findings are also consistent with prior findings that SOFA overestimates mortality among Black patients and underestimates mortality among White patients with sepsis and ARDS prior to the COVID-19 pandemic [29]. The existing literature suggests that both

**Table 4. Univariate and multivariate regression model results for factors associated with SOFA score within 24 hours ≥ 6.**

| Characteristic | Model 1- Combined Univariate (unadjusted) | | | | Model 1- Combined Multivariate | | | |
|---|---|---|---|---|---|---|---|---|
| | OR | 95% CI | | p-value | OR | 95% CI | | p-value |
| **Race/Ethnicity** | | | | | | | | |
| Hispanic | 0.78 | 0.58 | 1.06 | 0.1077 | 0.87 | 0.62 | 1.24 | 0.4501 |
| Black Non-Hispanic | 1.58 | 1.21 | 2.06 | 0.0008* | 1.49 | 1.11 | 1.99 | 0.0075* |
| White Non-Hispanic | reference | | | | Reference | | | |
| **Age** | | | | | | | | |
| 18–34 | reference | | | | Reference | | | |
| 35–64 | 2.55 | 1.44 | 4.52 | 0.0013* | 1.95 | 1.07 | 3.54 | 0.0288* |
| > = 65 | 2.84 | 1.62 | 4.98 | 0.0003* | 2.57 | 1.32 | 4.98 | 0.0052* |
| **Sex** | | | | | | | | |
| Men | 1.83 | 1.45 | 2.32 | < .0001* | 1.94 | 1.51 | 2.50 | < .0001* |
| Women | Reference | | | | Reference | | | |
| **Insurance status** | | | | | | | | |
| Private | Reference | | | | Reference | | | |
| Medicare | 1.53 | 1.09 | 2.14 | 0.0135* | 1.07 | 0.70 | 1.63 | 0.757 |
| Medicaid | 1.40 | 0.94 | 2.07 | 0.0955 | 1.35 | 0.89 | 2.04 | 0.1536 |
| Uninsured | 1.01 | 0.58 | 1.76 | 0.9678 | 1.19 | 0.66 | 2.13 | 0.566 |
| **BMI** | | | | | | | | |
| <25 | Reference | | | | Reference | | | |
| 25–29.9 | 1.07 | 0.79 | 1.45 | 0.6708 | 1.22 | 0.88 | 1.68 | 0.2317 |
| 30–34.9 | 1.08 | 0.77 | 1.52 | 0.6628 | 1.30 | 0.91 | 1.87 | 0.1561 |
| 35+ | 1.17 | 0.85 | 1.62 | 0.3372 | 1.52 | 1.07 | 2.18 | 0.0207* |
| **Comorbid conditions** | | | | | | | | |
| Chronic pulmonary disease | 1.27 | 0.99 | 1.62 | 0.0603 | 1.07 | 0.81 | 1.40 | 0.6453 |
| CHF | 1.86 | 1.45 | 2.38 | < .0001* | 1.19 | 0.86 | 1.63 | 0.288 |
| Diabetes | 1.65 | 1.31 | 2.08 | < .0001* | 1.07 | 0.82 | 1.41 | 0.6082 |
| CAD | 1.89 | 1.48 | 2.42 | < .0001* | 1.18 | 0.86 | 1.61 | 0.3151 |
| Hypertension | 1.67 | 1.27 | 2.19 | 0.0003* | 0.95 | 0.67 | 1.34 | 0.7594 |
| Advance renal disease | 3.15 | 2.29 | 4.34 | < .0001* | 2.35 | 1.62 | 3.40 | < .0001* |
| Advance liver disease | 2.96 | 1.58 | 5.54 | 0.0007* | 2.51 | 1.29 | 4.89 | 0.0068* |

Abbreviations

*: p-value < 0.05; BMI: Body Mass Index; CAD: Coronary Artery Disease; CHF: Congestive Heart Failure; CI: Confidence Interval; OR: Odds Ratio; SOFA: Sequential Organ Failure Assessment.

differences in individual characteristics (income, comorbid conditions) and hospital characteristics contribute to racial disparities in COVID-19 outcomes in the US [36].

The racial disparities in SOFA scores we found among patients with COVID might be due to the systemic overestimation of mortality among Black persons or underestimation of mortality among White persons. This might occur for example because Black patients at baseline have higher creatinine levels than White patients, leading to an elevation of the renal component of the SOFA score unrelated to any illness [37]. Alternatively, Black persons with COVID-19 might have higher SOFA scores in the hospital because COVID-19 affects them more severely, for example because they are subjected to higher levels of discrimination and stress or because they have less access to long-term preventive care, quality education, economic stability, and other social determinants of health [23, 24, 38]. Finally, Black patients might have higher SOFA scores at the time of admission because they present or are admitted

to hospitals only when they are sicker [39, 40]. This could occur because of current or prior discrimination within the healthcare system that might discourage patients from seeking medical attention with mild or moderate symptoms [41].

Of these potential contributing factors, prior or current anti-Black discrimination, leading to distrust of the healthcare system and delays in hospital admission, could explain our current findings. Of note, Non-Hispanic Black patients but neither Hispanic patients, patients with Medicaid, nor uninsured patients demonstrated a greater risk of an elevated SOFA score within 24-hours of admission. This would suggest an etiology, such as anti-Black stigma and resulting delays in hospital admission, that affects Black patients in the US to a greater degree than other marginalized populations such as patients with Medicaid or without insurance, who would also be expected to have reduced income, access to preventative care, and other social determinants of health. This potential mechanism accords with prior reports of Black patients in the US presenting to medical attention at more advanced stages of illness [39, 42]. This proposed mechanism is tentative because our analysis does not directly examine income or wealth but uses the imperfect proxy of Medicaid status. Further study, involving more detailed socioeconomic data will be necessary to support or refute this proposed mechanism.

Because published US triage protocols utilized the SOFA score to allocate scarce medical resources, and prioritized patients with lower SOFA scores over other patients, such protocols–if implemented–would be more likely to triage Non-Hispanic Black people to not receive scarce resources such as ventilators and ICU beds during future waves of the COVID-19 pandemic. As part of a system that predictably leads to racial disparities in health outcomes, triage protocols have the potential to become a component of systemic racism.

Given these findings and the possibility that crisis standards of care may be implemented during the future pandemics, it is important to prospectively consider and implement measures to reduce systemic racism, as well as socioeconomic and disability barriers to equal access to healthcare. The ideal would be to minimize or prevent entirely the need for triage, particularly among marginalized populations. This might be achieved in the short term through vaccination support, public health education, distribution of personal protective equipment, stockpiling of critical medical resources, targeted COVID-19 testing, contact tracing, social distancing, and even lockdowns coupled with financial support. The manifest injustice of the systemic racism and health inequities that COVID-19 has highlighted should also motivate long-term efforts to achieve more equitable health outcomes in the United States. These might include universal health insurance [43], a more redistributive system of taxation [44], housing support [45], elimination of food deserts and neighborhood segregation [46], anti-racism trainings for clinicians [47], robust collection of race and ethnicity health data [48], expanded access to higher and medical education, and recruitment of marginalized populations into the medical workforce [49]. Such interventions, while requiring large investments on a societal level, could reduce health disparities on a much broader and more lasting scale than interventions tailored specifically to the COVID-19 or other pandemics.

It is also possible to make crisis standards of care and triage protocols themselves more equitable. Such efforts would not be expected to reduce disparities beyond the setting of a pandemic, and then only for those patients who were subjected to rationing. The development, revision, and oversight of these protocols might be made more open and transparent to patients, community members, and to the general public. Healthcare systems and states might recruit triage advisory and oversight committees that specifically include robust representation from ethnic and racial minorities, as well as individuals with disabilities and other marginalized populations [7]. Committees might specifically recruit advocacy organizations, faith leaders, institutional diversity officers, and other community leaders to ensure adequate

representation of community concerns. The triage teams that implement protocols in hospitals might also be mandated to include representation of diverse perspectives.

In addition, the SOFA score might be supplemented to achieve more equitable outcomes. Prioritarian triage protocols might still use mortality probability scoring, such as the SOFA score, but might give marginalized populations a bonus or prioritization in these assessments. For example, patients might have their priority score improved slightly on the basis of their home address, using the Area Deprivation Index [50]. The impact of such modifications to SOFA scores on identified racial disparities would be an important area for future study. Potential comparative advantages and disadvantages of alternative triage systems are reviewed elsewhere [51].

Our study is limited in that it was conducted within a single healthcare system in the Northeastern United States. Our healthcare system experienced a surge of COVID patients relatively early in the pandemic, with a peak on April 22, 2020 followed by relatively lower numbers, and medical care for COVID-19 has evolved over the course of the pandemic. The disparities that this study documents may not be generalizable to other regions with different racial and ethnic demographics within the United States or globally. Our study is also limited by the relatively small number of patients with very elevated SOFA scores, which prevented analysis of more detailed SOFA score categories. Future research should include larger sample sizes which would allow for these types of analyses.

Perhaps most importantly, our study is limited by the data available within the clinical EMR. For example, racial and ethnic data is generally documented by unit clerks based on their observation of patients rather than on patient's self-identification. Patients classified as Hispanic often lack race data, which prevents us from identifying Hispanic Black vs. Hispanic White patients in our analyses. Prior studies have shown that "socially assigned" race does associate closely with health outcomes [52]. Another limitation is that we did not investigate potential disparities in SOFA scores in other marginalized populations. In particular, our EMR does not include socioeconomic data beyond insurance status, which is a poor surrogate for income, wealth, education, or occupation. Our insurance data are further limited because home care and disability data are not available. Evaluating the degree to which racial and ethnic disparities in the US, such as those identified in this study, are caused by or independent of socioeconomic inequality is a critical question that will require future study with detailed socioeconomic data [53]. Our study was also limited in the paucity of data available on some important clinical comorbidities. Future research in this area should include more detailed and complete data on clinical comorbidities that may affect mortality, such as dementia and cancer diagnoses. In addition, future research is needed to examine the effects of disability, psychiatric comorbidities, substance use disorders, unstable housing, or incarceration on SOFA scores.

In conclusion, Non-Hispanic Black patients admitted to hospitals with COVID-19 had increased odds of an elevated SOFA score $\geq 6$ within the first 24-hours of admission. Therefore, published triage protocols utilizing the SOFA score to allocate scarce medical resources would be more likely to deny Non-Hispanic Black patients scarce medical resources such as ventilators and ICU beds if implemented during the COVID-19 pandemic. Governments and healthcare systems should prospectively consider and implement measures to reduce systemic racism, protect marginalized populations, and promote racial and ethnic equity during pandemics.

## Acknowledgments

The authors wish to acknowledge the support of the Center for Medical Informatics and the Equity Research and Innovation Center at Yale School of Medicine. In particular we are

indebted to Indira Flores, and Olamide Olawoyin for assisting in literature review and project planning.

## Author Contributions

**Conceptualization:** Benjamin Tolchin, Carol Oladele, Nitu Kashyap, Mary Showstark, Michelle C. Salazar, Jennifer L. Herbst, Steve Martino, Nancy Kim, Katherine A. Nash, Max Jordan Nguemeni Tiako, Shireen Roy, Rebeca Vergara Greeno, Karen Jubanyik.

**Data curation:** Carol Oladele, Deron Galusha, Nitu Kashyap, Jennifer Bonito.

**Formal analysis:** Benjamin Tolchin, Carol Oladele, Deron Galusha, Nitu Kashyap, Jennifer L. Herbst, Nancy Kim, Karen Jubanyik.

**Investigation:** Benjamin Tolchin, Carol Oladele, Nitu Kashyap, Mary Showstark, Michelle C. Salazar, Jennifer L. Herbst, Steve Martino, Nancy Kim, Katherine A. Nash, Max Jordan Nguemeni Tiako, Shireen Roy, Karen Jubanyik.

**Methodology:** Benjamin Tolchin, Carol Oladele, Deron Galusha, Nitu Kashyap, Mary Showstark, Jennifer Bonito, Michelle C. Salazar, Jennifer L. Herbst, Steve Martino, Nancy Kim, Karen Jubanyik.

**Project administration:** Jennifer Bonito, Rebeca Vergara Greeno.

**Supervision:** Benjamin Tolchin, Carol Oladele, Steve Martino, Karen Jubanyik.

**Validation:** Benjamin Tolchin, Carol Oladele.

**Writing – original draft:** Benjamin Tolchin, Carol Oladele.

**Writing – review & editing:** Benjamin Tolchin, Carol Oladele, Deron Galusha, Nitu Kashyap, Mary Showstark, Michelle C. Salazar, Jennifer L. Herbst, Steve Martino, Nancy Kim, Katherine A. Nash, Max Jordan Nguemeni Tiako, Shireen Roy, Rebeca Vergara Greeno, Karen Jubanyik.

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
