## [Decision Letter · Decision Letter 0]

11 May 2021

PONE-D-21-10700

Racial Disparities in the SOFA Score Among Patients Hospitalized with COVID-19

PLOS ONE

Dear Dr. Tolchin,

Thank you for submitting your manuscript to PLOS ONE. After careful consideration, we feel that it has merit but does not fully meet PLOS ONE’s publication criteria as it currently stands. Therefore, we invite you to submit a revised version of the manuscript that addresses the points raised during the review process.

Please address all concerns raised by our reviewers, including details in the separated document.

Regarding an expression in statistical analysis, multivariable regression models should be used, rather than multivariate; the former would treat multiple factors, the latter multiple outcomes in the models.

In addition, further analysis should be needed to validate the study findings, as one of reviewers pointed out.

We look forward to receiving your revised manuscript.

Kind regards,

Takeru Abe, Ph.D

Academic Editor

PLOS ONE

Journal Requirements:

2. Please include your statement regarding patient consent in the manuscript Methods.

3.We note that you have indicated that data from this study are available upon request. PLOS only allows data to be available upon request if there are legal or ethical restrictions on sharing data publicly. For information on unacceptable data access restrictions, please see http://journals.plos.org/plosone/s/data-availability#loc-unacceptable-data-access-restrictions.

Reviewers' comments:

Reviewer's Responses to Questions

**Comments to the Author**

1. Is the manuscript technically sound, and do the data support the conclusions?

Reviewer #1: Yes

Reviewer #2: Partly

2. Has the statistical analysis been performed appropriately and rigorously? 

Reviewer #1: Yes

Reviewer #2: No

3. Have the authors made all data underlying the findings in their manuscript fully available?

Reviewer #1: No

Reviewer #2: No

4. Is the manuscript presented in an intelligible fashion and written in standard English?

Reviewer #1: Yes

Reviewer #2: Yes

5. Review Comments to the Author

Reviewer #1: The author is to be congratulated for submitting this manuscript. The following are the comments:

1. In the current COVID-19 pandemic, this manuscript is ever so important for some countries in which certain residents are more seriously prone to contracting COVID-19. Also, certain ethnic groups have seen both certain morbidity and mortality disparities.

2. In this manuscript, the author reported that non-Hispanic blacks in the United States have significantly higher SOFA scores at upon hospital arrival. Thus, possibly resulting in lesser opportunities to receive critical life-saving medical resources. Yet, the author did not answer the reasons for this phenomenon.

3. One of this manuscript’s major concerns was to evaluate the relationship between ethnicity and systemic racism in the clinical scope. Yet, the author did not separate the Hispanic Americans into Hispanic-Black and other ethnic groups. Thus, causing potential challenges in evaluating the impacts of different skin complexions in certain racism-related issues.

4. The SOFA score is a validated tool to identify patients who have lesser survival opportunities. SOFA should only be used in certain extreme critical situations such as disaster-related ones. Per the author's suggestion, a bonus or prioritization in assessment might be given to certain marginalized populations. This revision of prioritizing triage protocols can only be used in the short term. In the future, the correction of potential systemic racism factors in the clinical scope should be achieved. Detailed discussions about the pro and cons of any protocol revisions or policy change to achieve equality should thus be provided.

5. A stratified study per different ethnic groups is extremely important in medical racism studies. Thus, the supplementary S1 Table “Characteristics of COVID+ patients by Race/Ethnicity” should be moved to the results section. Although the results might not be significant, the data in each cell can provide important information for the readers. A detailed discussion of this table is recommended.

6. Please provide the demographic data of the population serviced by the YNHH, especially those involving the ethnicity, occupation, and other socioeconomic status-related factors.

Reviewer #2: The topic is important, and the research idea is outstanding. Given the relevancy of the topic, major revision should be done to achieve a more informative paper. In particular: more complete information about SOFA and triaging should be available in the background. A multilevel analysis should be included to better explore the social factors, or hospital-level factors, contributing to disparity.

Please refer to the attached document.

6. PLOS authors have the option to publish the peer review history of their article (what does this mean?). If published, this will include your full peer review and any attached files.

Reviewer #1: No

Reviewer #2: **Yes: **Andrea Rossi Zadra

---

## [Author Response · Author response to Decision Letter 0]

29 Jul 2021

We are deeply grateful for the reviewers thoughtful and detailed comments. Please see our detailed uploaded response to reviewers' and editors' comments.

---

## [Decision Letter · Decision Letter 1]

31 Aug 2021

PONE-D-21-10700R1

Racial Disparities in the SOFA Score Among Patients Hospitalized with COVID-19

PLOS ONE

Dear Dr. Tolchin,

Thank you for submitting your manuscript to PLOS ONE. After careful consideration, we feel that it has merit but does not fully meet PLOS ONE’s publication criteria as it currently stands. Therefore, we invite you to submit a revised version of the manuscript that addresses the points raised during the review process.

We look forward to receiving your revised manuscript.

Kind regards,

Takeru Abe, Ph.D

Academic Editor

PLOS ONE

Journal Requirements:

Additional Editor Comments:

Thank you for submitting your revision to the PLOS ONE.

Most of the previous comments have been addressed.

There is one comment from our Reviewer 1, as described below, which would be important to address.

Please respond to it adequately.

Reviewers' comments:

Reviewer's Responses to Questions

**Comments to the Author**

1. If the authors have adequately addressed your comments raised in a previous round of review and you feel that this manuscript is now acceptable for publication, you may indicate that here to bypass the “Comments to the Author” section, enter your conflict of interest statement in the “Confidential to Editor” section, and submit your "Accept" recommendation.

Reviewer #1: (No Response)

Reviewer #2: All comments have been addressed

2. Is the manuscript technically sound, and do the data support the conclusions?

Reviewer #1: Yes

Reviewer #2: Yes

3. Has the statistical analysis been performed appropriately and rigorously? 

Reviewer #1: Yes

Reviewer #2: Yes

4. Have the authors made all data underlying the findings in their manuscript fully available?

Reviewer #1: Yes

Reviewer #2: No

5. Is the manuscript presented in an intelligible fashion and written in standard English?

Reviewer #1: Yes

Reviewer #2: Yes

6. Review Comments to the Author

Reviewer #1: Table 2 (Characteristics of COVID+ patients by Race/Ethnicity) showed the vulnerability of Non-Hispanic Black patients in their health and socioeconomic status, which may lead to higher SOFA scores. Since this manuscript intended to explore the racial disparities in SOFA Score, the significance of this table is worth mentioning in the discussion section.

Reviewer #2: I wish to congratulate with the outstanding work, and I appreciate the revision of the paper.

I have been pleased from reviewing this outstanding work and I support its publication on PLOS One.

7. PLOS authors have the option to publish the peer review history of their article (what does this mean?). If published, this will include your full peer review and any attached files.

Reviewer #1: No

Reviewer #2: **Yes: **Andrea Rossi Zadra

---

## [Author Response · Author response to Decision Letter 1]

31 Aug 2021

Please see the uploaded revision notes for detailed response to the requested revision. We are deeply grateful to the editors and reviewers for their thoughtful feedback.

---

## [Decision Letter · Decision Letter 2]

6 Sep 2021

Racial Disparities in the SOFA Score Among Patients Hospitalized with COVID-19

PONE-D-21-10700R2

Dear Dr. Tolchin,

We’re pleased to inform you that your manuscript has been judged scientifically suitable for publication and will be formally accepted for publication once it meets all outstanding technical requirements.

Kind regards,

Takeru Abe, Ph.D

Academic Editor

PLOS ONE

Additional Editor Comments (optional):

Reviewers' comments:

Reviewer's Responses to Questions

**Comments to the Author**

1. If the authors have adequately addressed your comments raised in a previous round of review and you feel that this manuscript is now acceptable for publication, you may indicate that here to bypass the “Comments to the Author” section, enter your conflict of interest statement in the “Confidential to Editor” section, and submit your "Accept" recommendation.

Reviewer #1: All comments have been addressed

2. Is the manuscript technically sound, and do the data support the conclusions?

Reviewer #1: Yes

3. Has the statistical analysis been performed appropriately and rigorously? 

Reviewer #1: Yes

4. Have the authors made all data underlying the findings in their manuscript fully available?

Reviewer #1: Yes

5. Is the manuscript presented in an intelligible fashion and written in standard English?

Reviewer #1: Yes

6. Review Comments to the Author

Reviewer #1: The author is to be congratulated for this interesting and meaningful manuscript. The last comment has been fully addressed. I appreciate the revision and believe that this manuscript is suitable to be published on PLOS One.

7. PLOS authors have the option to publish the peer review history of their article (what does this mean?). If published, this will include your full peer review and any attached files.

Reviewer #1: No

---

## [Editor Report · Acceptance letter]

9 Sep 2021

PONE-D-21-10700R2 

Racial Disparities in the SOFA Score Among Patients Hospitalized with COVID-19 

Dear Dr. Tolchin:

I'm pleased to inform you that your manuscript has been deemed suitable for publication in PLOS ONE. Congratulations! Your manuscript is now with our production department. 

Kind regards, 

on behalf of

Dr. Takeru Abe 

Academic Editor

PLOS ONE